# A 30-month Field Evaluation of Low-Cost $CO_2$ Sensors Using a Reference Instrument

Qixiang Cai[1,2,8], Ning Zeng[3*], Xiaoyu Yang[4], Chi Xu[5], Zhaojun Wang[4], Pengfei Han[6,7*]

[1]State Key Laboratory of Numerical Modeling for Atmospheric Sciences and Geophysical Fluid Dynamics, Institute of Atmospheric Physics, Chinese Academy of Sciences, Beijing 100029, China

[2]Key Laboratory of Urban Meteorology, China Meteorological Administration, Beijing 100089, China

[3]Department of Atmospheric and Oceanic Science, and Earth System Science Interdisciplinary Center, University of Maryland, College Park, Maryland 20742, USA

[4]Shandong Jinan Ecological and Environmental Monitoring Center, Jinan 250102, China

[5]State Environmental Protection Key Laboratory of Quality Control in Environmental Monitoring, China National Environmental Monitoring Centre, Bejing 100012, China

[6]State Key Laboratory of Atmospheric Environment and Extreme Meteorology, Institute of Atmospheric Physics, Chinese Academy of Sciences, Beijing 100029, China

[7]Carbon Neutrality Research Center, Institute of Atmospheric Physics, Chinese Academy of Sciences, Beijing 100029, China

[8]Qiluzhongke Institute of Carbon Neutrality, Jinan 250100, China

*Correspondence to*: Pengfei Han (pfhan@mail.iap.ac.cn); Ning Zeng (zeng@umd.edu)

**Abstract.** Low-cost, medium-precision sensor (LCSs) networks have emerged as a promising approach for $CO_2$ monitoring under complex urban emission conditions. However, the field performance of these LCSs faces significant challenges from environmental factors (e.g., temperature, humidity) and long-term drifts caused by sensor degradation (e.g., light source aging). In this study, we performed 30 months of co-located observations using LCS units (named SENSE-IAP) alongside a Picarro reference analyzer to evaluate long-term field performance, which is essential for the correction and validation of mid-low cost $CO_2$ observation networks. The environmental correction system we developed effectively corrected the impact of daily environmental changes, which reduced the root mean square errors (RMSE) from $5.9 \pm 1.2$ ppm to $1.6 \pm 0.5$ ppm for SENSE-IAP. The corrections remained robust against seasonal environmental changes, maintaining daily RMSE typically within 1-3 ppm throughout the 30 months of observation. Long-term drifts, commonly occurred in LCS, produced biases of up to 27.9 ppm over two years. Furthermore, seasonal drift cycle contributed up to 25 ppm RMSE after six months of deployment. Although the environmental correction system could not address these errors, linear interpolation effectively calibrated the long-term drift, reducing the 30-month RMSE to $2.4 \pm 0.2$ ppm. To improve the accuracy of high-density $CO_2$ networks using such LCSs, we recommend maintaining a calibration frequency preferably within three months and not exceeding six months, with optimal calibration performed during both winter and summer to ensure accuracy within 5 ppm. These findings indicate that SENSE-IAP instruments can operate long-term without requiring return to the laboratory for re-calibration or frequent field standard gas calibration, thereby substantially reducing time, labor, and financial costs.

Keywords: low-cost sensors; environmental correction; long-term drift calibration; $CO_2$ monitoring network

## 1. Introduction

Urban $CO_2$ emissions demonstrate complex spatial and temporal variability (Wada et al., 2011), influenced by diverse emission sources (Gurney et al., 2012; Kellett et al., 2013), meteorological factors (Grimmond et al., 2002; Lateb et al., 2016) and potential misinterpretation from biogenic fluxes (Miles et al., 2021). These complexities pose significant challenges in accurately capturing and interpreting the intricate changes in urban $CO_2$ concentrations, highlighting the necessity for more advanced and comprehensive monitoring solutions.

Recent advancements in medium-precision carbon monitoring technologies have enabled the establishment of high-density sensor networks with low-to-mid cost and medium precision, presenting a viable and competitive monitoring strategy (Müller et al., 2020; Shusterman et al., 2018). This approach effectively addresses the challenges posed by $CO_2$ variability in urban environments, where complex emission sources, vegetative carbon sinks, and dynamic meteorological conditions interact. Unlike high-precision instruments (0.1 ppm) such as Picarro cavity ring-down spectroscopy (CRDS) analyzers (Picarro, 2023), which cost approximately $100,000, low-cost ($30-$500) and mid-cost ($1,000-$5,000) sensors utilizing non-dispersive infrared (NDIR) technology (Table 1) (Han et al., 2025), offer a 1-10 ppm accuracy at a fraction of the price (30-5000 USD) - representing a cost reduction of more than an order of magnitude. This enhanced affordability facilitates large-scale deployment, making these sensors particularly attractive for comprehensive urban $CO_2$ monitoring systems (Lopez-Coto et al., 2017; Turner et al., 2016; Wu et al., 2016; Zeng et al., 2021).

Table 1.   Types of low-cost and mid-cost sensors.

| Sensor type | Typical cost range (US$) | Typical precision (ppm $CO_2$) | $CO_2$ sensors | Network |
|---|---|---|---|---|
| **Low-cost** | $30 - $500 | 1-10 ppm | SenseAir LP8; SenseAir K30 | Switzerland, Carbosense $CO_2$ sensor network (Müller et al., 2020) Beijing, SENSE-JJJ (Han et al., 2024) |
| **Mid-cost** | $1,000 - $5,000 | 1 - 4 ppm | Vaisala GMP343; SenseAir HPP | California, BEACO$_2$N (Asimow et al., 2024) Switzerland, ZiCOS-M (Grange et al., 2025) Paris, $CO_2$ sensor Network (Lian et al., 2024) |

While cost-effective, NDIR sensors are sensitive to environmental changes and often exhibit long-term drifts and abrupt jumps. Noise, environmental sensitivity, and temporal drifts result in raw measurements that typically have more significant errors and uncertainties than the accuracy and resolution required for urban $CO_2$ monitoring. The accuracy of

these sensors typically depends on correction methods that account for environmental factors such as temperature, humidity, and pressure. However, their sensitivities of these variables can vary significantly, presenting major challenges in calibrating large sensor networks (Bigi et al., 2018; Delaria et al., 2021; Hagan et al., 2018; Martin et al., 2017). With careful correction—using environmental chambers and co-located high-precision instruments—these sensors can achieve short-term measurement accuracy within ± 1-3 ppm under controlled laboratory conditions (Cai et al., 2024; Grange et al., 2024; Müller et al., 2020).

Additionally, unlike high-precision instruments, NDIR sensors are more prone to temporal drift and fluctuations. To mitigate these issues, sensors are periodically re-calibrated the laboratory or undergo in-situ field calibration using traceable standard gases. While essential, these correction processes are labor-intensive and time-consuming. Some networks adopt alternative approaches. For instance, sensor-specific drift slope can be predetermined before deployment, with offsets corrected using the lower percentile (5-10%) of observations across the entire network (Shusterman et al., 2016). Delaria et al. (2021) demonstrated that using the median value from at least 12 sites with minimal temperature dependence as a reference can maintain network precision at 3.6 ppm. Another method involves determining drift by comparing nearby instruments under specific weather conditions when horizontal $CO_2$ gradients are small (Müller et al., 2020). A widely used and robust calibration method involves automatic or manual standard gas injections. In the Beijing and Jinan networks, we applied this method across more than 160 instruments, with automatic standard gas calibration performed weekly. Results showed mean biases of -1.28 ~ -0.64 ppm at a one-month scale (Cai et al., 2024). Manual standard gas calibration is particularly useful in mobile observations, such as on-road vehicle measurements and vertical profile observations using tethered balloons (Liu et al., 2021; Bao et al., 2020). However, calibration costs can be significant. For automatic standard gas calibration (weekly frequency), expenses reach approximately $300 per station per year, consuming two 8L, 10 MPa gas tanks (one as a working standard and the other for quality control). Consequently, maintaining 100 stations with such low-cost sensors would incur an annual cost of $30,000 for standard gas alone.

Currently, several cities such as California, Paris, Switzerland and Beijing have established high-density $CO_2$ monitoring networks utilizing NDIR sensors (Table 1). For instance, the Berkeley Environmental Air-quality and $CO_2$ Network (BEACO$_2$N) in California, USA, utilizes the Vaisala CarboCap GMP343 sensor (mid-cost), with a raw accuracy of ± 3ppm+1% reading (Vaisala, 2020). After correcting for bias and temporal drift using the in-situ method, the observation accuracy is approximately 1-4 ppm (Shusterman et al., 2016, 2018), while the reported accuracy improved to 1.6-3.6 ppm after temperature correction (Delaria et al., 2021). The Carbosense $CO_2$ sensor network in Switzerland (Müller et al., 2020), which uses the SenseAir LP8 sensor (low-cost), with a raw accuracy of ± 50 ppm (SenseAir: LP8 Product Sheet, 2019),

achieves an observation accuracy of 8-12 ppm through initial laboratory chamber correction and regular drift calibration via ambient co-location with nearby reference instruments.

China aims at peaking carbon emissions before 2030 and achieving carbon neutrality before 2060 (the Dual Carbon Goals, DCGs) (He et al., 2020; Huang et al., 2023; Zeng et al., 2022). To support China's DCGs and address the high spatial variability of $CO_2$ concentration in urban areas, the Institute of Atmospheric Physics, Chinese Academy of Sciences (IAP), established a network of 134 sites using SenseAir K30 sensor since 2017 (Han et al., 2024). This study presents the correction methods developed for the SenseAir K30 sensors and evaluates the accuracy achievable through the environmental dependence correction method based on laboratory simulation. At a field observation site in Beijing, environmentally corrected low-cost sensors (LCSs) were co-located with high-precision Picarro instruments for up to 30 months. By comparing the data with measurements from Picarro instruments, we gained more profound insights into the long-term performance and long-term drift characteristics of the LCS, as well as assessed the effectiveness of our long-term drift correction method. Our findings demonstrate that timely long-term drift correction significantly improved the accuracy of urban $CO_2$ monitoring networks based on LCS and reduced time, labor, and money investment. This research provides valuable evidence for optimizing the deployment and maintenance of LCS-based monitoring networks in urban environments.

## 2. The application of SenseAir K30 sensors for urban $CO_2$ monitoring

A multivariate linear regression analysis was used for environmental correction, which can improve the accuracy of the SenseAir K30 sensor from its initial specification of ± 30 ppm ± 3 % of reading (SenseAir: K30 products sheets, 2022) to a range of 1.7-4.3 ppm (Martin et al., 2017). The environmentally corrected K30 sensor demonstrated reliability and consistency when compared to higher-precision instruments and standard gas under a controlled indoor environment, with a root mean square error (RMSE) ranging from 1 to 3 ppm on a monthly scale (Cai et al., 2024). Furthermore, the low-cost sensor exhibited highly consistent with Picarro system during on-road observations conducted using the same vehicle, with an RMSE of 3.6 ppm (Liu et al., 2021). In a study by Bao et al., (2020), the sensor was utilized to measure the $CO_2$ vertical profile in the lower troposphere in Hebei Province, China, showing good consistency with traditional gas chromatography measurements (Bao et al., 2020). Additionally, Cai et al., (2024) applied the low-cost sensor in an industrial park, revealing a $CO_2$ concentration enhancement of 5-28 ppm within the park compared to a reference site (Cai et al., 2025).

The Beijing-Tianjin-Hebei (or Jing-Jin-Ji, JJJ) network, deployed with low-cost sensors, has provided valuable insights into seasonal variations, urban-rural differences, and the homology of $CO_2$ and $PM_{2.5}$ (Han et al., 2024). The low-cost

sensors have also been proven effective in detecting signals related to COVID-19. Continued $CO_2$ measurements in Beijing showed a 15-ppm reduction during the 2020 lockdown period compared to the before and after periods. Similarly, regular on-road $CO_2$ observation in Beijing before, during, and after COVID-19 lockdown showed a 40–60ppm decrease during COVID-19 lockdown period (Liu et al., 2021). These applications demonstrate the versatility and reliability of low-cost sensors in capturing both environmental and anthropogenic influences on atmospheric $CO_2$ concentrations.

**3. Instrument design and correction methods of SENSE-IAP**

The SENSE-IAP instrument integrates three K30 sensors alongside a Bosch BME680 (BME) sensor (Bao et al., 2020; Liu et al., 2021) , all collected by an updated version of BeagleBone Green Wireless (BBGW). The standard version of SENSE-IAP instrument also includes a Figaro TGS 2611 sensor for $CH_4$ detection and a Plantower PMSA003 for $PM_{2.5}$ measurements. These components are compactly integrated onto a single circuit board and housed within a weatherproof

enclosure, as illustrated in Fig. 1.

The BME sensor is positioned close to the K30s to simultaneously monitor the temperature (T in °C), relative humidity (RH in %), and pressure (P in hPa) of the air mass inside the instrument. This design ensures real-time correction of the sensor response values, accounting for dynamically changing external environmental conditions and enhancing measurement accuracy.

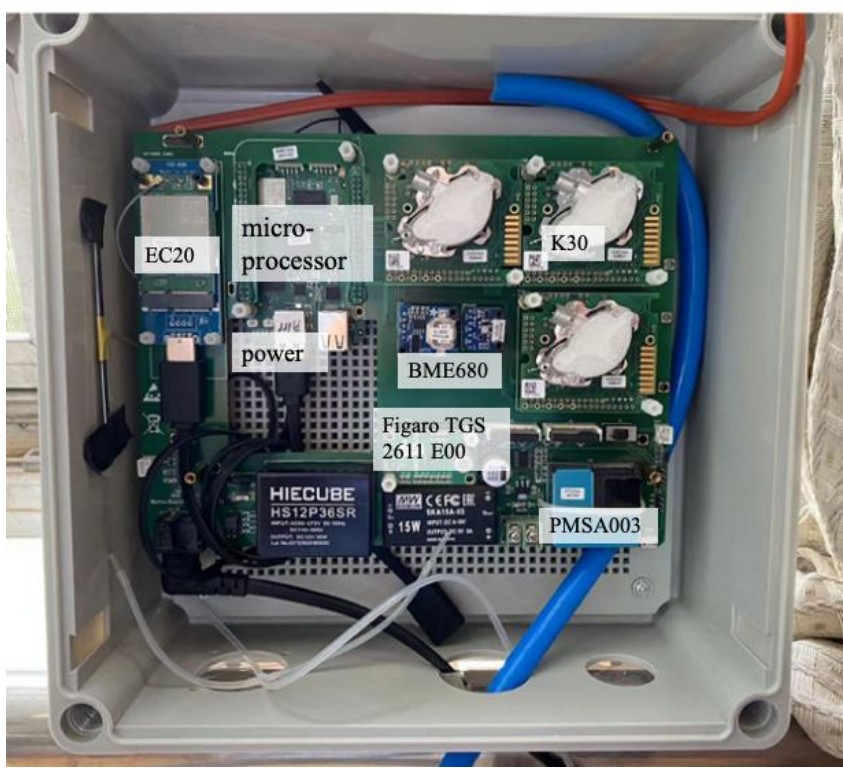

**Figure 1.** The layout of sensors in the standard version of SENSE-IAP instrument.

To improve the observing accuracy, we developed a $CO_2$ calibration system incorporating both controlled environmental experiments and calibration software (Bao et al., 2020; Han et al., 2024; Liu et al., 2021). The system follows

these main steps: 1) Converting raw electrical signals to $CO_2$ values within atmospheric observation ranges; 2) Resampling data from 2-second to 1-minute to reduce white noise; 3) Applying environmental corrections of temperature, humidity and air pressure during span calibration; 4) Performing calibration using standard gas traceable to the WMO X2007 scale (identical to the standard used in high-precision systems like Picarro instruments, to remove system bias. The controlled environmental experiments typically require one week of co-location with a reference instrument to determine environmental correction coefficients, followed by another week of co-location for post-quality check. To fully characterize long-term drift patterns, we recommended maintaining observations for at least one year to capture seasonal variations (including both summer and winter cycles).

The raw signals from all sensors were collected at a frequency of 2 seconds, with a standard deviation of approximately ± 4 ppm. Fig. 2 shows the experimental results of continuously introducing standard gas over 25 hours to evaluate the instrument's noise characteristics. As shown in Fig. 2, the Allan deviation (in ppm) decreases with increasing integration time. At a 2-second measurement interval, the noise level is 4 ppm, which decreases to approximately 0.2 ppm for integration times ranging from 2 minutes to 1 hour. However, the Allan deviation increases after 1 hour of integration time, indicating the presence of drift contributions.

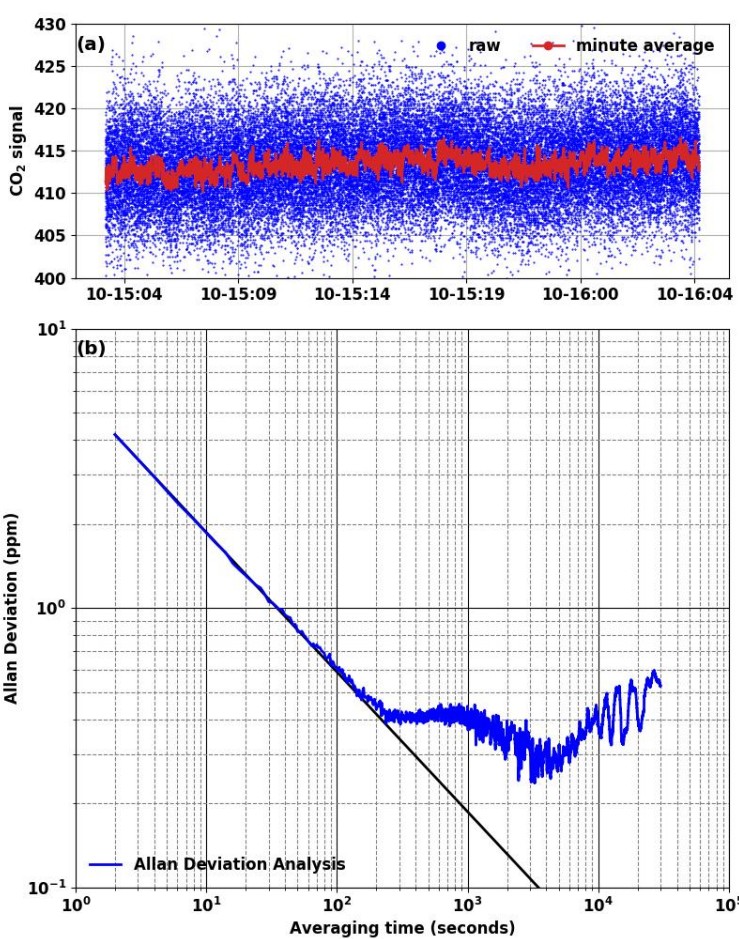

**Figure 2.** (a) The raw signals measured continuously over 25 hours from 03:00 on October 15th to 03:00 on October 16th, 2022. (b) Allan deviation log plots.

In addition to white noise, the raw signals often contain outliers that must be removed through quality control. According to formula 1, following the 3-sigma principle, the original data points xi collected during the sampling period are treated as samples. The average value $\bar{x}$ of non-missing $x_i$ is calculated, along with the standard deviation (SD) of the xi is σ. If | xi-$\bar{x}$|>4 σ, the data point is identified as an outlier and removed. We adopted the 4σ threshold to strike a balance between effectively removing outliers and preserving the natural ranges of variability in the data.

$$\sigma = \sqrt{\frac{\sum_{i=1}^{n}(x_i - \bar{x})^2}{n-1}} \quad (1)$$

Subsequently, the raw signals are averaged from a 2-second interval to a 1-minute interval (resample) to reduce standard deviation. A 1-minute integration time was chosen as an optimal trade-off between noise reduction and maintaining sufficient time resolution to track natural variations in $CO_2$ concentrations accurately. As formula 2, the resampled value $Y_i$ for each minute is calculated as the average of all non-outlier values $x_j$ within that minute.

$$Y_i = \frac{\sum_{j=1}^{m} x_j}{m} \quad (2)$$

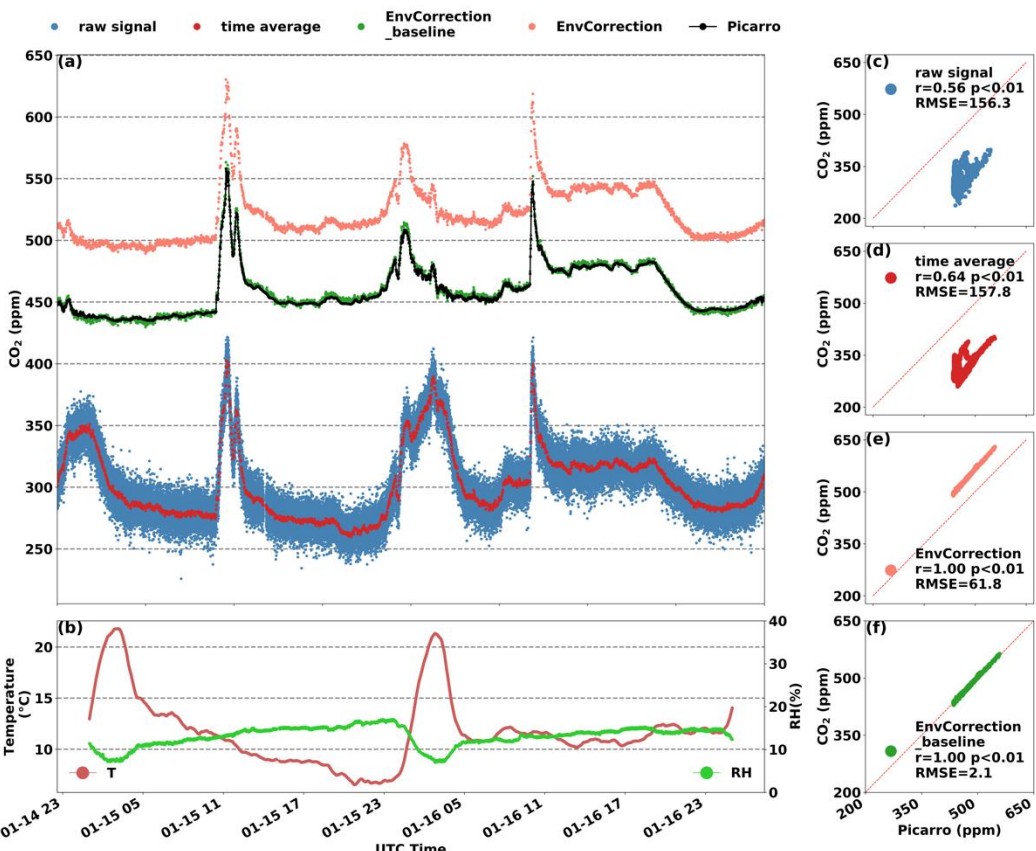

**Figure 3.** (a) Compared the $CO_2$ at each processing step, (b) Synchronized monitoring of T and RH. (c-f) The scatter plots of $CO_2$ measurements form LCS and Picarro instrument at different processing steps, including (c) the raw signal in 2-second resolution, (d) the values after noise reduction at a 1-minute resolution, (e) the $CO_2$ concentrations after environment corrections and (f) systematic bias correction.

The correction system we developed substantially improves the accuracy of $CO_2$ measurements through a comprehensive process that includes outlier removal and noise reduction. Fig. 3 shows the main steps of the correction system, with the data cleaning method described earlier constituting the initial two steps. As shown in Fig.3 (a), the raw signal (blue) undergoes de-specking and denoising (red). However, a noticeable deviation remains between the LCS measurements and the true value, with a correlation coefficient (r-value) of approximately 0.6 (Fig.3(c-d)). The differences mainly come from environmental sensitivity and baseline deviations in concentration.

Similar to the correction standards used in high-precision systems, our LCS units are calibrated using standard gas traceable to the WMO X2007 scale. This calibration adjusts the span and calibrates system bias before deployment. For the typical $CO_2$ concentration range (400-700ppm), a concentration-dependent offset ($\Delta C$) exists between the time-averaged LCS measurements and the standard gas concentration. Since this concentration dependence varies for each K30 sensor, laboratory corrections are essential for accuracy improvement. Our concentration correction process includes multiple concentration gradients, ensuring applicability to real-world monitoring scenarios. The fitting parameters of $\Delta C$ against the measured values are determined through regression analysis, enabling precise correction across the operational range.

$$\Delta C = Y_m - y_0 \quad (3)$$

where, $y_0$ represents the concentration of standard gas or the high-precision reference instrument (Picarro G2301); $Y_m$ is the minute-averaged values from LCSs.

Environmental calibration was conducted in an environmental controlled chamber. The LCSs environmental sensitivity corrections included T compensation (10-50 °C in 5 steps), RH compensation (10% -90% in 9 steps), and P compensation. This correction is based on sensitivity testing conducted in the laboratory (Martin et al., 2017), with comparisons made against Picarro. Each sensor is assigned unique sensitivity parameters through multivariate regression and iteration analysis.

$$\Delta C = f(a_T, Y_T) + f(a_H, Y_H) + f(a_P, Y_P) + f(a_C, Y_C) + \varepsilon \quad (4)$$

where, the baseline correction coefficient is $\varepsilon$; the $Y_T$, $Y_H$, $Y_P$, and $Y_C$ represent the compensation values for the T, RH, P, and concentration sensitivity, respectively, applied to the minute-averaged $CO_2$ measurement; the $a_T$, $a_H$, and $a_P$ are the regression coefficients against T, RH and P respectively, while $a_C$ is the span correction coefficient against concentration.

Thus, the corrected $CO_2$ can be expressed as:

$$C = Y_m - (f(a_T, Y_T) + f(a_P, Y_P) + f(a_H, Y_H) + f(a_C, Y_C) + \varepsilon) \quad (5)$$

The r values between the $CO_2$ corrected in the final two steps and the Picarro measurements are close to 1 (Fig. 3(e-f)). Additionally, the difference between the environmental correction (pink) and baseline calibration (green) in Fig. 3(a) represents the coefficient $\varepsilon$. After applying these correction steps, the accuracy of the LCS measurements improved to 1-4 ppm compared to Picarro (Liu et al., 2021).

Before deployment, the span and system bias of the LCSs were calibrated. However, once deployed to field stations, LCSs tended to drift on a weekly to monthly scale, necessitating time-dependent drift calibration. We defined $S_{cor}$ as the starting time of drift and $E_{cor}$ as the time when the drift slope stabilized or when calibration was required, with $\Delta = E_{cor}\text{-}S_{cor}$. $\Delta C_{drift}$ represents the bias between the concentration $C$ measured by the instrument and the standard concentration $C_0$ at $E_{cor}$. Using the formula (6), the drift rate over time (ppm/min) at $E_{cor}$ is calculated as $s_t$. The $b_{cal}$ is a constant deviation, representing the difference between the baseline and the standard value before long-term drift occurs (at $S_{cor}$). This value is generally considered zero, since system bias has been calibrated before departure. The error at any time between $S_{cor}$ to $E_{cor}$ can be calibrated to $C_{Cor}^{drift}$ using the following formulas:

$$\Delta C_{drift} = s_t \Delta t + b_{cal} \qquad (6)$$

$$C_{Cor}^{drift} = C - \Delta C_{drift} \qquad (7)$$

This integrated instrument with environmental correction and drift correction is named as SENSE-IAP.

## 3. Co-located observation system

Our experiment has been conducted since July 2022 at IAP (Beijing-IAP site, Fig. 4a and 4b). Located in a central urban area with high population density, the Beijing-IAP site is significantly influenced by traffic emissions.

To evaluate the performance of LCS, we developed a synchronous observation system that compares LCS with high-precision instruments. This system includes two SENSE-IAP units (numbered pi688 and pi736), each equipped with three K30 sensors. A cavity ring-down spectrometer (Picarro G2301) was used as the high-precision instrument for $CO_2$ measurements (Picarro, 2023). The precision and accuracy of the Picarro instrument are better than 0.1 ppm (Yang et al., 2021). At the Beijing-IAP, the Picarro analyzer was calibrated monthly using high-pressure standard gases provided by the Meteorological Observation Center of the China Meteorological Administration (MOC/CMA), which are traceable to the World Meteorological Organization (WMO) X2007 scale.

To ensure long-term synchronous observation between the LCSs and Picarro, deployment enables two sets of instruments measure the same gas mass. This ensures that any differences in observed values only come from the effects of T, RH and P as well as the concentration span, all of which can be adjusted through correction methods. The deployment setup is shown in Fig. 4(b-c).

The instruments were mounted on the edge of an open window to directly measure the outdoor air and environmental changes, which were almost the same as field-deployed conditions (Fig. 4 and S8). Temperature primarily ranged from 0-40°C, and RH typically between 0-60%. A partition isolated the LCSs from indoor spaces, ensuring measurements

primarily reflected outdoor environmental conditions. Three K30 sensors and one BME sensor were housed in a transparent

cover featuring two 2-mm-diameter ventilation holes at the upper-left and lower-left corners (red circles in Fig.4 c and d).

Air passively diffuses into the pi688 enclosure, while a 4-mm-diameter blue tube connected to the cover's left side enabled

active air sampling via an air pump (GAST DOA-P504-BN). Upstream of the pump, the air flow passed through a

capsule-type10-μm filter (COBETTER 92WM-LPF1000) for particulate matter removal (Fig.4 c). Downstream of the pump,

a Nafion drying tube is installed for Picarro analysis (bypassed for SENSE-IAP measurements) (Fig. 4c). The filtered air is

then divided by a four-way valve, delivering 3-5 L/min to the pi736 and 0.3-0.4 L/min to the Picarro analyzer, with excess

air vented through the final outlet.

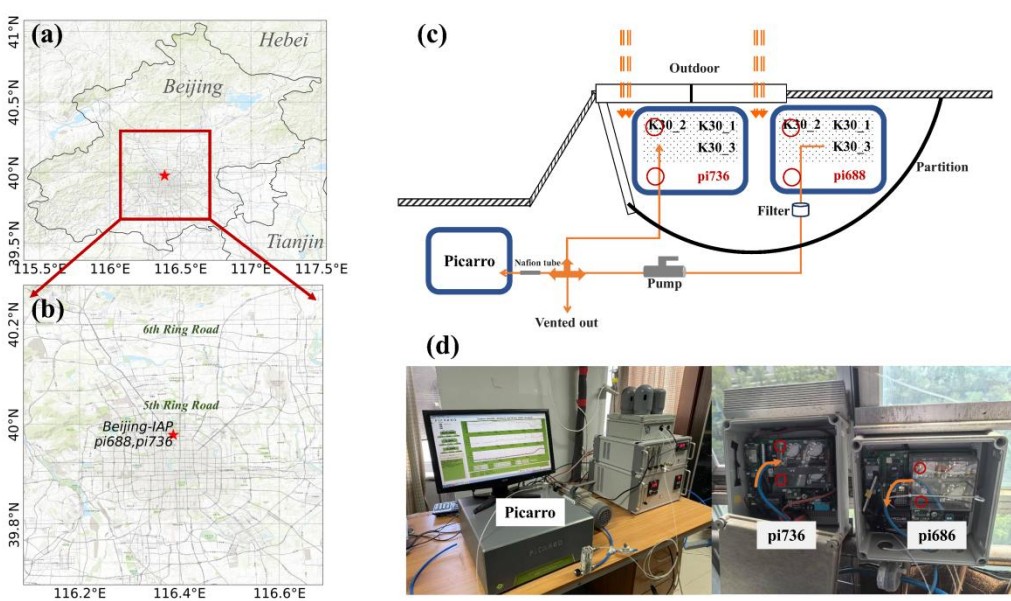

**Figure 4**. (a) Map of the location of Beijing-IAP in Beijing, (b) Map showing the location of Beijing-IAP within Beijing's

main urban area. (c) the diagram of gas flow design for the synchronous observation system, (d) the photographs of the

instrument installation setup. The source for the basemap used in subplot (a-b) was from ESRI

(https://server.arcgisonline.com/arcgis/rest/services/World_Topo_Map/MapServer).

## 4. Environmental corrections for field measurements

To distinguish between the effects of short-term environmental factors and long-term drifts, we present these

corrections separately. To focus specifically on the environmental corrections, all results in this section were obtained after

removing long-term drift. A comprehensive discussion of long-term drift is provided in Section 5. Fig. 5 shows the results

from environment-corrected SENSE-IAP at the Beijing- IAP site, compared with those from the Picarro system. After

approximately two weeks of data collection during both summer and winter, the SENSE-IAP showed highly consistent

results with Picarro, with RMSEs of 1.6 ppm in summer and 1.8 ppm in winter. In contrast, the raw $CO_2$ concentration data

from the SenseAir showed a higher RMSE of 6.2 ppm in summer and 7.0 ppm in winter. Furthermore, the effectiveness of

environmental correction is evident across different seasons.

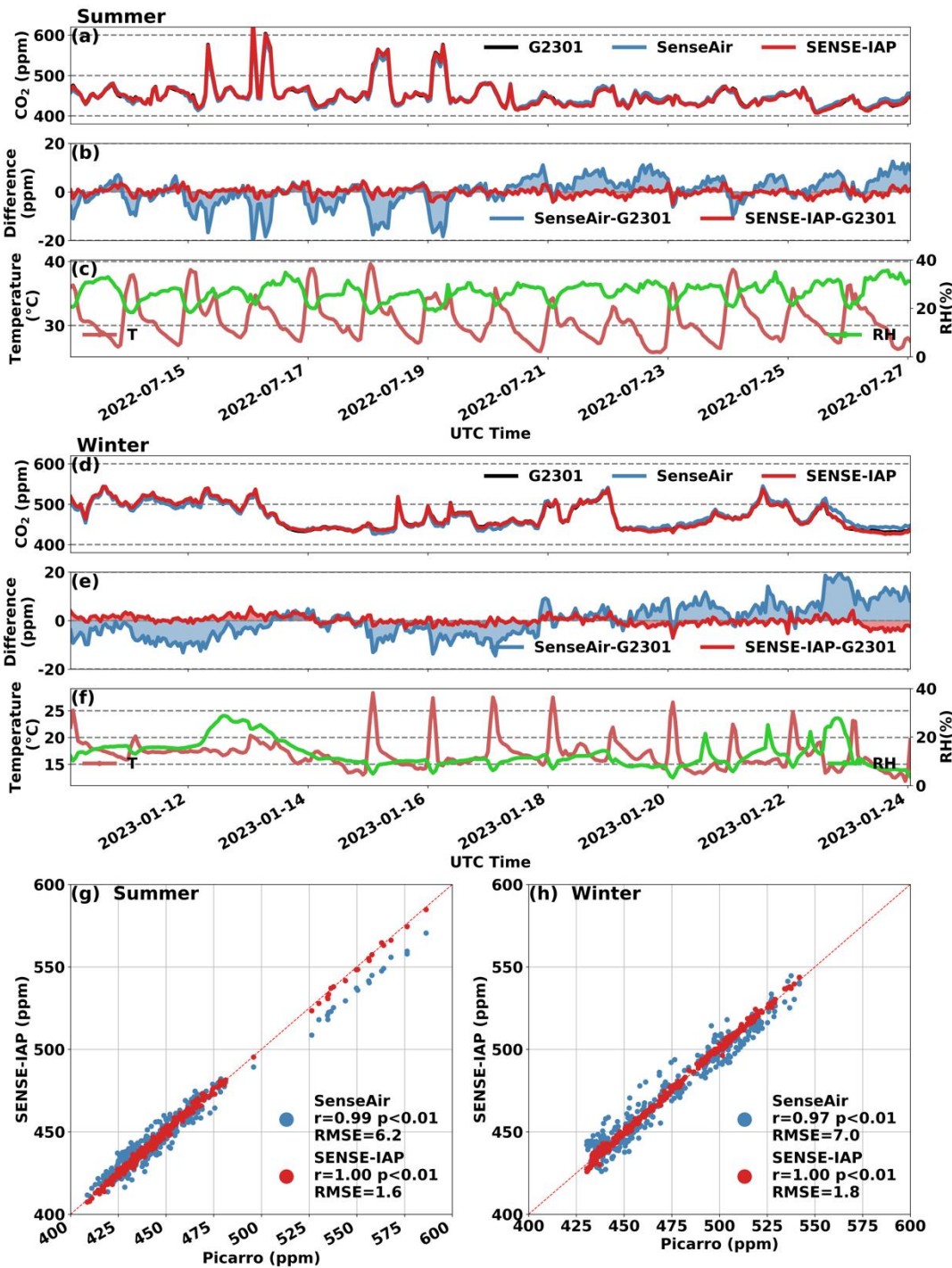

**Figure 5**: Comparison between hourly $CO_2$ concentrations measured by the SENSE-IAP and Picarro systems at the Beijing

site from July 13[th] to 27[th] in 2022 (a-c, g) and January 10[th] to 24[th] in 2023 (d-f, h). We compared both the raw $CO_2$ data from

SenseAir (blue) and the environmentally corrected data from SENSE-IAP (red) with that from Picarro (black). The T and

RH were measured for the environment inside the instrument.

The correction system effectively adjusted the $CO_2$ concentration within the 400-700 ppm measurement range. As shown in Fig. 5b, during the period from July 16th to 19th, even when the ambient $CO_2$ concentration experienced significant fluctuations, the instrument showed high consistency with Picarro. The T and RH detected by the BME sensor were used to monitor the instrument's internal environment. Notably, in winter mornings, sunrise caused a significant temperature increase due to the presence of metal components on the circuit board (Fig. 5f). Our environmental correction successfully corrected for the temperature dependence, as the deviation between the SenseAir and Picarro showed a strongly correlated with temperature in both seasons (Fig. S1a). Additionally, the deviation of SenseAir relative to Picarro was significantly associated with RH in summer (Fig. S1b), and our correction system incorporated humidity, which was related to ambient temperature. Compared to the raw SenseAir data, the consistency of all six sensors improved markedly, with the RMSE decreasing from $5.0 \pm 1.0$ ppm to $1.3 \pm 0.2$ ppm in summer and from $6.8 \pm 0.8$ ppm to $2.0 \pm 0.4$ ppm in winter (Fig. S2).

Our correction system can perform environmental sensitivity analysis and correction on individual sensors, with the correction efficacy remaining robust across seasonal environmental changes. We further analyzed the daily RMSE of the SENSE-IAP relative to Picarro during 30 months of co-located observation. As shown in Fig. 6a, the daily RMSE of one sensor (pi688-K30) ranged from 1.5 to 4.0 ppm throughout the observation period, with the light blue shadow representing the monthly mean ± standard deviation. The average and median of daily RMSEs for this sensor were less than 2.0 ppm (Fig. 6b), except for an increase to higher than 2 ppm in summer (dark green). Compared to Picarro, the consistency of SENSE-IAP was better in winter, with an RMSE below 2.0 ppm on most days. Except for a few sensors exhibiting slightly higher RMSEs (approximately 3.0 ppm) in spring and autumn (pi732-K30), the daily RMSEs of the six sensors showed no significant seasonal variation (Fig. 6b).

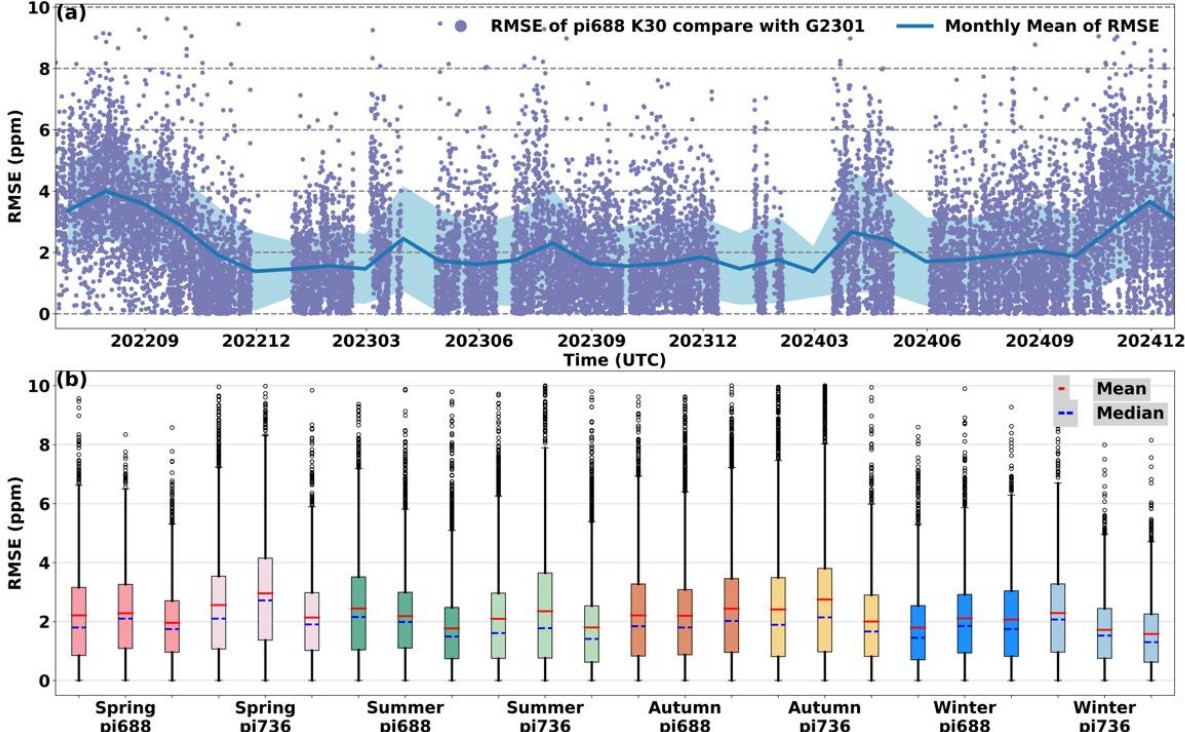

**Figure 6.** (a) The time series of daily RMSEs for hourly $CO_2$ concentration relative to Picarro (purple points) from June 2022 to Dec 2024, with a monthly rolling mean (blue line and shadow). (b) Box plot of the daily RMSEs of all six sensors across different seasons. Sensors from the same instrument are represented in the same colors (spring: dark pink/pink; summer: dark green/green; autumn: orange/yellow; and winter: dark blue/blue). Within each box, the red line indicates the mean value, while the blue dashed line represents the median values.

## 5. Performance of typical long-term drift and correction method

After environmental correction, the six SENSE-IAP sensors were co-located with a Picarro analyzer synchronously for over 30 months. As shown in Fig. 7, two types of long-term drift were identified: 1) a downward drift trend and 2) a seasonal drift cycle. While the environmental correction system effectively corrects the impact of diurnal environmental changes (Fig. 6), significant errors occurred in sensors due to the two types of long-term drift. Without the long-term drift calibration algorithm, the bias of SENSE-IAP could reach 27.9 ppm, with an RMSE of approximately 28.1 ppm (Fig. 7).

During the observation period, all six sensors exhibited long-term downward drifts ranging from 0.1 to 1.2 ppm per month (ppm/mo) (Fig. 7 and Table 2). Among the six sensors deployed in this study, only pi736 K30_3 showed a drift trend of less than 0.1 ppm/mo (Table 2). For sensors such as pi688-K30 and pi688-K30_2, the $\Delta CO_2$ displayed a continuous downward trend over the 30 months, with slopes of 1.2 and 1.0 ppm/mo, respectively (Table 2). Notably, the drift trends for these sensors did not show significantly stabilization over time.

In addition to the downward drift trend, sensors like pi688-K30_3 and pi736-K30 exhibited varying seasonal cycle trends. After six months of deployment, these sensors showed RMSEs of 25.3 and 24.8, respectively (Fig. 7). However, after more than one year of observation, the impact of seasonal drift decreased, and the errors caused by long-term drift were highlighted, resulting in reduced RMSEs as 17.3 and 10 ppm, respectively. The RMSE evaluation was conducted during the identical period employed for drift correction.

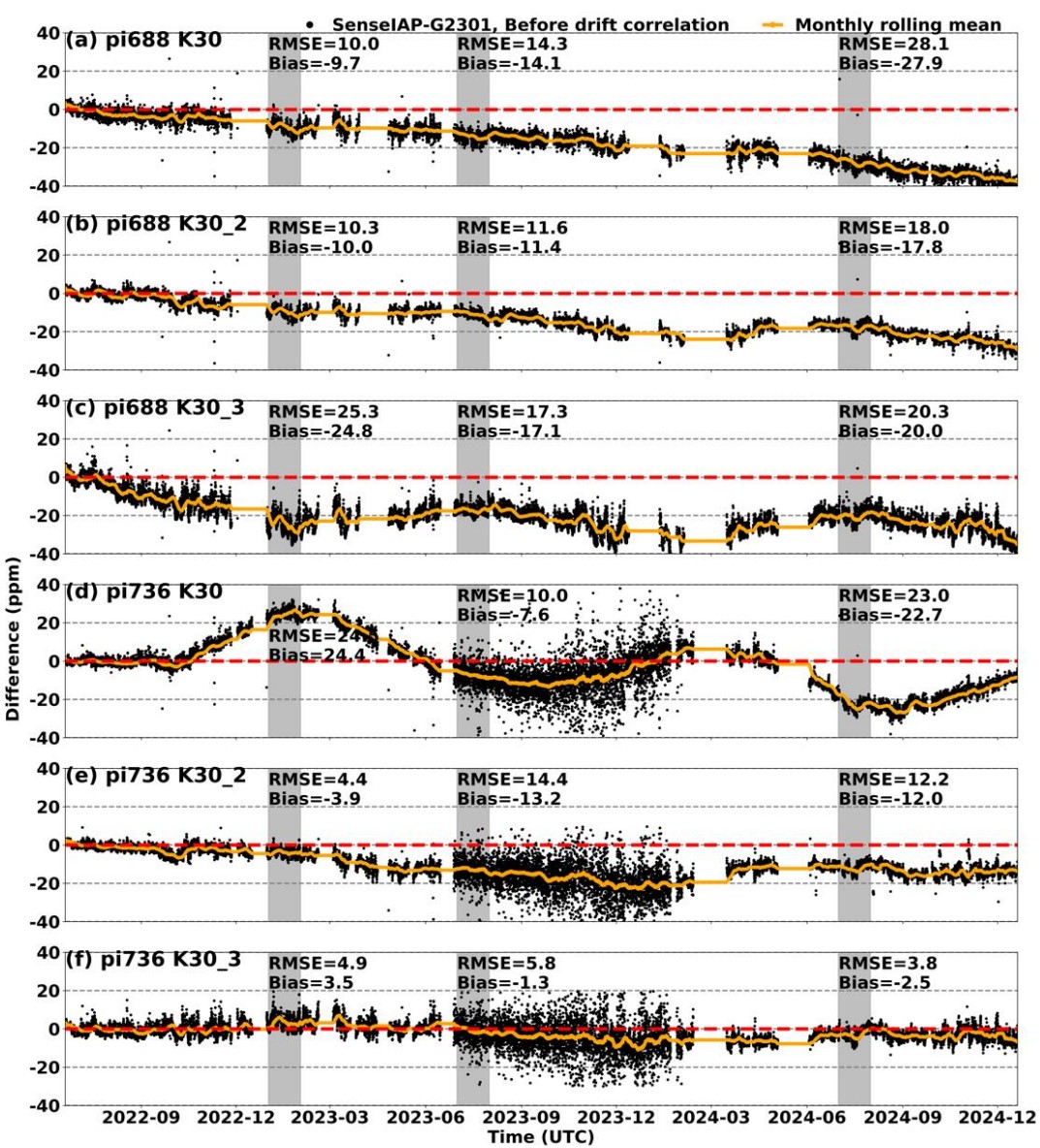

**Figure 7**: Time series of $\Delta CO_2$ for the six sensors at the Beijing-IAP from June 2022 to Dec 2024 (black), with the monthly rolling mean (yellow line). The gray shadow represents the one-month time range used to evaluate the RMSE and bias for each sensor after half a year, one year, and two years of deployment.

From the perspective of drift magnitude, a significant bias of 5 ppm (approximately 1% of the ambient $CO_2$ concentration) typically occurred within 3-10 months after calibration, with most cases observed within 5 months. The

seasonal drift cycle occurred on a six-month scale, with maximum errors typically occurring in winter and summer. Therefore, we recommend that the long-term drift calibration frequency of SENSE-IAP should be no less than three months and no longer than six months. In addition, drift calibration should be performed at least once during both winter and summer seasons. If the target monitoring accuracy is within 3 ppm, the drift calibration frequency should be at least every two months, as a 3-ppm bias typically develops within 2-5 months.

Table 2. The Long-term drift trend of six sensors (Unit: ppm/mo)

| SENSE-IAP | pi688 | | | pi736 | | |
|---|---|---|---|---|---|---|
| Slope (ppm/mo) | s1 | s2 | s3 | s1 | s2 | s3 |
| drift in the first year | -1.2 | -1.0 | -1.4 | -0.6 | -1.1 | -0.1 |
| drift in the second year | -1.2 | -0.5 | -0.2 | -1.3 | 0.1 | -0.1 |

The method for the long-term drift calibration is as follows. According to functions 6-7, we illustrate our long-term drift calibration method by focusing on the first year of observations. At the start of the observation period (Jun 2022), we adjusted the baseline of the six sensors. We designated the initial calibration time point as $S_{cor}$ for the first observation period (Jun 2022 to Jan 2023). The inflection point of the drift trend in Feb 2023 was identified as $E_{cor}$ for the first period and as $S_{cor}$ for the second period of the observation (Jan 2023 to Sep 2023). The $E_{cor}$ for the second period was set to Sep 2023 in this study. We applied the linear calibration method between these two time points for both periods of the drift trend. After calibration, the $CO_2$ concentrations from the six sensors showed strong consistency with the Picarro, with an RMSE ranging from 2.4-3.0 ppm (Fig. 8).

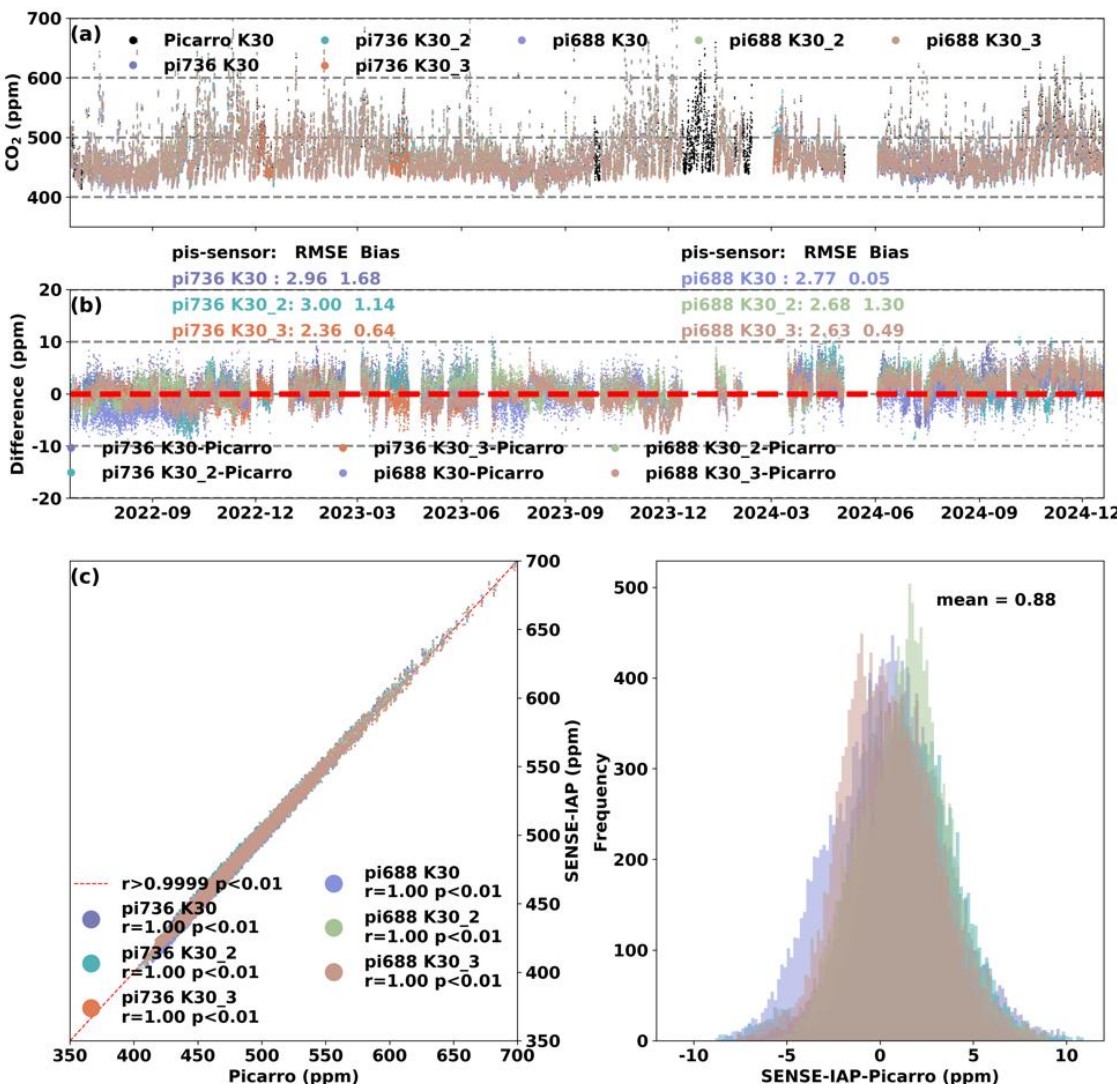

**Figure 8**: (a) Comparison of hourly $CO_2$ concentrations measured by six sensors and Picarro at Beijing- IAP from June 2022 to Dec 2024; (b) the time series of $\Delta CO_2$; (c) scatter plot of SENSE-IAP and Picarro; (d) histogram plot of the $\Delta CO_2$.

It should be noted that due to the loose connection of ventilation pipeline between Jul 2023 to Mar 2024, the $CO_2$ concentration measured by pi736 and Picarro were not strictly synchronized. This issue led to a relatively lower short-term monitoring accuracy during this period, primarily due to the lag effect caused by air diffusion. To assess the actual hourly monitoring accuracy of SENSE-IAP, we excluded data from this period. Without this exclusion, the RMSE for pi736 would have been 3.9-4.5 ppm (Fig. S3). However, to ensure the completeness and robustness of the long-term drift analysis, we

retained samples from this period, allowing for a more comprehensive evaluation of drift trends over time.

## 6. Comparison of data quality across multiple levels over a long-term scale

As previously mentioned, the correction system effectively corrected the sensors' short-term environmental dependence. By applying a linear calibration method at least every three months, we can calibrate the time-dependent drift of LCSs and

resolve residual seasonal cycles that the environmental correction system cannot fully resolve. This seasonal variation is prominently reflected in the original electrical signal (raw signal). As shown in Figure S4(b), we eliminated the influence of short-term environment changes through a 24-hour running mean, revealing that the seasonal drift cycle in the raw signal can reach up to 100ppm, with an RMSE of 38.2 and a bias of -21.1 ppm (Table 3).

The defining characteristic of medium-precision low-cost sensors is the presence of long-term drift. This drift, which exhibits a downward trend, is observed in all sensors, with variations only in the drift rate (Fig.S5). Long-term drifts resulted in an RMSE of 15.8 and a bias of -12.0 for the LCS, respectively (Table 3). Although seasonal cycle does not significantly alter the overall trend of long-term drift, seasonal environmental variations can introduce errors of up to 25 ppm if baseline calibration are performed only annually (Fig. 7). Therefore, we recommend that baseline calibration be conducted at least every six months, ideally during both winter and summer.

The data provided by the SenseAir manufacturer were corrected for temperature sensitivity using the default temperature parameters. The long-term drift was calibrated using a so-called ABS algorithm, which employs periodic one-point calibration in SenseAir, assuming that the minimum value of $CO_2$ concentration is 400 ppm in fresh air (SenseAir-Corrected data). However, due to the constant assumption of fresh air concentration, coupled with the carbon absorption of vegetation in summer and the higher emissions in winter, the SenseAir-Corrected data exhibit a fluctuating trend of approximately 20 ppm, with higher values in summer and lower values in winter. Consequently, the bias of SenseAir-Corrected data can be -3.3 ± 1.4 ppm, with an RMSE of 12.1 ± 2.0 ppm (Fig. S6, Table 3). In contrast, the drift-calibrated SENSE-IAP data demonstrate a much smaller bias (0.8 ± 0.4 ppm) and an 80% improvement in accuracy, with an RMSE of 2.4 ± 0.2 ppm (Fig.S7, Table 3).

**Table 3**: Evaluation parameters of the $CO_2$ concentration measured by K30 sensors compared to those from Picarro, including SenseAir-Corrected values, Raw signal, and SENSE-IAP at the Beijing-IAP from June 2022 to December 2024 (unit: ppm).

| Data Type | SenseAir-Corrected | | Raw signal | | SENSE-IAP-Env-Corrected | | SENSE-IAP-Env+Drift-Corrected | |
|---|---|---|---|---|---|---|---|---|
| Sensors | RMSE | Bias | RMSE | Bias | RMSE | Bias | RMSE | Bias |
| pi688 K30 | 15.9 | -5.7 | 34.4 | -20.2 | 20.1 | -16.9 | 2.2 | 0.1 |
| pi688 K30_2 | 10.1 | -3.1 | 35.1 | -19.0 | 15.7 | -13.6 | 2.4 | 1.3 |
| pi688 K30_3 | 10.2 | -3.3 | 45.2 | -24.2 | 21.5 | -20.0 | 2.3 | 0.5 |
| pi736 K30 | 10.8 | -1.1 | 22.0 | -11.8 | 13.5 | -3.5 | 2.6 | 1.1 |
| pi736 K30_2 | 13.0 | -2.6 | 38.4 | -23.0 | 14.3 | -10.8 | 2.8 | 1.0 |
| pi736 K30_3 | 12.5 | -4.2 | 53.9 | -28.1 | 9.8 | -7.0 | 2.2 | 0.7 |
| Mean | 12.1 | -3.3 | 38.2 | -21.1 | 15.8 | -12.0 | 2.4 | 0.8 |
| SD | 2.0 | 1.4 | 9.9 | -5.1 | 4.0 | 5.6 | 0.2 | 0.4 |

*According to the statistical results of 24-hour running means.

## 7. Seasonal drift cycle effects on SENSE-IAP

As shown in Fig. S8, the seasonal variations observed before instrument linear calibration were correlated with T, RH and P. The $\Delta CO_2$ between pi736-K30 and Picarro significantly correlated with all three environmental factors, with r values of -0.58, -0.46, and 0.33 against T, RH and P, respectively. In contrast, the relationship between pi688-K30_3 and environmental factors was opposite to that of pi736-K30, with r values of 0.33, 0.5, and -0.6 against T, RH and P, respectively. Considering seasonal phase differences between $CO_2$ concentration changes and environmental factors, this seasonal deviation was likely attributable not only to insufficient environmental compensation but also to the influence of seasonal effects on the instrument's physical properties. For instance, changes in the sensor's optical cavity size caused by thermal expansion and cold contraction can change the optical path-lengths, subsequently affecting the pressure within the optical cavity and the strength of infrared $CO_2$ absorption (Yao et al., 2023). However, this hypothesis cannot fully explain why the two sensors, pi688 K30_3 and pi736 K30, exhibited opposite drift directions during the same season.

Long-term drift is typically observed in low- and mid-cost NDIR $CO_2$ sensors. Although our findings provide valuable references data specifically for K30 sensors, the recommended calibration frequency and observed seasonal cycle characteristics may also apply to other similar NDIR sensors. The calibration method we employed is universally applicable and effective for long-term drift corrections. For other NDIR sensors, we recommend selecting several representative samples for co-location with high-precision instrument under field conditions for a minimum of one year to fully

characterize their performance. This extended co-location period is crucial for comprehensively for evaluating both long-term drifts trends and seasonal drift patterns. Such characterization studies provide essential guidance for implementing remote calibration in high-density sensors networks. Alternatively, standard gas calibration serves as a practical substitute method. Based on our experimental results, we recommend a calibration frequency of at least once every one to three months for this approach.

## 8. Conclusions

We evaluated low-cost NDIR $CO_2$ sensors using Picarro as a reference instrument. Our environmental correction system effectively corrected the impact of short-term daily environmental changes by assigning unique environmental sensitivity parameters to each sensor. This approach reduces the short-term RMSE from 5.9 ± 1.2 ppm for SenseAir to 1.6 ± 0.5 ppm for SENSE-IAP. The correction system demonstrates robustness against seasonal environmental variations, maintaining a daily RMSE of 1-3 ppm.

Based on a 30-month observation, we recommend that the calibration frequency for long-term drifts not exceed six months. For optimal performance and to ensure the target monitoring accuracy remains within 1% of the ambient $CO_2$ concentration, a three-month calibration interval is recommended. If standard instruments, standard gases (which are generally easier to obtain), or other reliable concentration references such as model simulations are available, long-term drift can be linearly corrected at the seasonal scale.

Consequently, after deployment, even with significant environmental changes around the instrument, there is no need to frequently bring the instruments back to the laboratory for re-correction of environmental impacts. After the long-term drift calibration, the RMSE of SENSE-IAP remains 2.4 ± 0.2 ppm even after 30 months of operation. This performance enables long-term deployment of the instruments, significantly reducing the maintenance costs associated with LCS.

**Supporting information**

The supplementary information is available online.

**Data availability statement**

The data used to generate the figures in this manuscript are available at https://doi.org/10.6084/m9.figshare.29310890.v3.

**Competing interests**

The authors declare that they have no conflicts of interest.

**Funding**

This research was supported by the National Key R&D Program of China (No. 2023YFC3705500 and 2017YFB0504000); Jinan Carbon Monitoring and Evaluation Pilot Project (grant no. SDGP370100000202202001740); the Qiluzhongke Institute

of Carbon Neutrality Program of Jinan Dual Carbon Simulator; and the CAS Proof of Concept Program: Carbon

Neutrality-oriented Urban Carbon Monitoring System and Its Industrialization (grant no. CAS-GNYZ-2022).

**Author contributions**

NZ, PFH and QXC conceived and designed the study. QXC and PFH collected and analyzed the datasets. XYY, CX, and ZJW discussed the sensor results. QXC led the writing of the paper with contributions from all the coauthors. All coauthors contributed to the descriptions and discussions of the manuscript.

**Acknowledgments**

We thank Mr. Cory Martin, Mr. Di Liu, Mr. Yinan Wang, Mr. Ming Cui, Mr. Jin Guo, Mr. Zhimin Zhang, and Mr. Yang Zi for their help in the SENSE-IAP instrument development, calibration and deployments.

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
