# Peer review of "A 30-month Field Evaluation of Low-Cost CO2 Sensors Using a Reference Instrument"

_EGUsphere, 2025_

## Author Response (AR1)

**Referee 1:**

**Overall good research and worthwhile of publication with revisions. Further details needed about the experiment setup and how correction coefficients were established. Given the current details it would be difficult for a reader to replicate this work. Additionally, minor spelling and language changes are needed.**

**Despite these notes, having a 30-month analysis alongside a CRDS for low-cost sensors is impressive, significant, and worth publication. I was excited to see this work, and long time-frame pieces like this can further the research community.**

Response: We thank you for your understanding and appreciation of this paper. We have revised the grammar and expression issues suggested by you, and updated the co-located system diagram and related explanations. We have provided detailed responses to some comments regarding the experimental setup and result analyses.

We provided all the Picarro and SENSE-IAP data used in this paper (publicly available at https://doi.org/10.6084/m9.figshare.29310890.v3). We added more descriptions on the calibration algorithm and procedures in the main text (lines 135-141), and these descriptions were also provided in several published paper (Bao et al., 2020; Han et al., 2024; Liu et al., 2021; Martin et al., 2017), and similar methods and precision were achieved in other studies (Lian et al., 2024; Shusterman et al., 2016). However, due to the commercial confidential requirements (Beijing, Jinan, Fuzhou, and several other cities' LCS networks we served), the detailed calculation formulas cannot be disclosed in this research paper. We hope these revisions and explanations can satisfy the reviewer.

**Comments:**

**Line 49: When referencing Picarro or ABB-LGR, might want to reference measurement techniques those instruments use (CRDS, etc.)**

Response: Thank you for your comment. We have add the techniques of the sensors.

**Line 50: Awkward phrasing for cost comparison, might want to use a direct cost range instead**

Response: We modified the section of introduction. We added a table (Table 1) to clearly compare the cost, accuracy, and deployment status (including key cities with dense networks) of different low- and mid-cost $CO_2$ sensors.

**Line 58: Which cities in particular are being referenced?**

Response: We added the cities such as Paris, California, Zurich, and Beijing in lines 65-89 (Table 1).

**Line 76: "deployed to co-locate" is awkward, should this be "colocated" instead?**

Response: Thank you for this advice, we revised accordingly.

**Line 83: Section 2 title is awkward - application to what?**

Response: Thank you for your comment. We added descriptions for clarity.
The title of section 2 is: "The application of SenseAir K30 sensors for urban $CO_2$ monitoring"

**Line 87: references "higher-quality" while majority of paper references higher precision**

Response: Thank you for your comment. Using 'precision' would be more appropriate. We have revised.

**Line 94: What is the JJJ network? This is the only time it is mentioned**

Response: Sorry for this abbreviation, which means the Beijing-Tianjin-Hebei area, short for Jing-Jin-Ji (JJJ), and we added the full name of 'JJJ'.

**Line 102: Check capitalization for Bao**

Response: The capitalization issues have been fixed.

**Lines 107 - 110: How does air move in the box? Is it passive? Later experiment mentioned a pump used to move air into the box but nothing mentioned here about that. Is the system designed to be used passively or will it have to be used with a pump when deployed in the field?**

Response: Thank you for this question. We added more descriptions in the main text. The SENSE-IAP instrument can be deployed for both active and passive ways, and we tested the passive way in this study, and the active way was also tested in a previous study which showed similar results. Specifically, ambient air was firstly passively diffused into three K30 sensors in the SENSE-IAP instrument (pi688), then a 4 mm-diameter Teflon tube connected to the pi688 allows an air pump to actively draw ambient air from the inside of this pi688 instrument box. The air flow is then split by a three-way valve, delivering a 5 L/min flow of air to both the inside of pi736 box and the Picarro analyzer.
We have improved the diagram of gas flow for the observation system in Figure 4 and provided more detailed descriptions of the instrument's ventilation method in lines 225-232.

**Lines 154 - 157: More details are needed about how the sensitivity correction is done - a reader can't replicate a similar experiment based on what is provided here. Was this done with an environmental chamber or on lab bench sampling**

ambient conditions? How long was this comparison done for, and at what temperature steps?

Response: Thank you for this question, and we added more descriptions in lines 183-184 and 135-141. Environmental calibration was done in an environmental controlled chamber. The procedure includes temperature correction, which is set at 5 steps from 10-50°C, and humidity correction, which is set at 9 steps from 10%-90% RH. The calibration is based on sensitivity test results obtained from this controlled environmental chamber. Through nonlinear fitting and iterative processing, unique parameters are derived for each individual sensor. From the controlled environmental experiments to post-stage precision and accuracy check, the sensors undergo continuous monitoring and is co-located with a Picarro 2401 analyzer.

Lines 193 - 196: More details are also needed about how the experiment works. Does the SENSE-IAP system have it's own pump? Is the outflow from the main pump being directed into the SENSE-IAP systems and the Picarro siphoning off them?

Response: We have improved the diagram of gas flow design for the observation system in Figure 4 and provided a more detailed description of the instrument's air flow and    ventilation in the last paragraph in section 3.

Line 194: What kind of dryer is being used?

Response: A capsule filter (COBETTER 92WM-LPF1000) is used to remove particulate matter, and a Nafion drying tube is used to remove moisture for Picarro use, and for the SENSE-IAP instrument, moisture is not removed. The relevant information has been revised in the main text (line 229-230).

Figure 4b: This figure is confusing - Which is the pump in the figure and where is it located? Also which is the 4-way valve?

Response: We have improved the diagram of gas flow design for the observation system in Figure 4c. There are two pumps in the system, one is the main pump before the 4-way valve, the other is a Picarro pump only used for Picarro air flow control.

Figure 6: This figure is good but I found the order confusing. Figure 6 is showing corrected results while 7 shows uncorrected results. It was difficult for me to follow the analysis with the shuffling back and forth.

Response: Thank you for this question, and we added more explanations at the beginning of Section 4 and 5. Figure 6 presents the results of short-term calibration, statistically presenting the daily consistency between the LCSs and Picarro analyzer throughout the entire observation period. These results illustrate both the efficacy of environmental sensitivity calibration and the stability of its corrective effects. And in

order to only focus on checking the environmental corrections, we removed long-term drifts here and discussed this important issue separately in the following section. Figure 7 illustrates the long-term drift characteristics of the LCSs. The data presented in Figure 7 have undergone environmental sensitivity calibration but remain uncorrected for the long-term drift.

**Line 238: "with [a] Picarro"**
Response: Thank you. We revised.

**Line 240: "effectively corrects the impacts of diurnal environmental changes" - need to back this up with either a figure or analysis from a previous section**
Response: Thank you for this comment. Lines 265-272, 282 and Figure 6 demonstrate the effectiveness of the calibration methodology for diurnal environmental changes.

**Lines 267 - 273: It was not clear to me if the RMSE evaluation was done on the same period used to do the drift corrections or another independent dataset.**
Response: It was done on the same period for drift correction and we clarified it in line 293-294. Lines 310-317 elaborates on the specific methodology we employed for drift correction using functions 6-7.

**Line 296: "manufacturer"**
Response: Thank you and revised.
**Line 364: Capitalization of names**
Response: Thank you and revised.

**Referee 2:**
**This is an interesting evaluation of the SenseAir K30 sensors, co-located with Picarros, over a 30 month timescale. This is a valuable study, but the manuscript needs significant revision to properly categorize sensor types and adjust all comparisons accordingly, address scalability concerns, strengthen the literature context, and better acknowledge limitations. It should be accepted for publication in AMT if the following can be addressed.**
**The authors fail to distinguish between "low-cost" and "mid-cost" $CO_2$ sensors. The Vaisala CarboCap GMP 343 sensor mentioned in line 59 and used in networks like ZICOS-M (https://acp.copernicus.org/articles/25/2781/2025/) and BEACO2N is at a significantly higher price point compared to the SenseAir K30 used in this paper and the Carbocaps are now usually called a "mid-cost" sensor to distinguish them for the LCS like SENSE-IAP. Please adjust the introduction to distinguish between sensors in the $10s-$100s USD (low-cost),**

**sensors in the mid-cost range ($1000s), and reference grade sensors (typically $10,000s).**

**Additionally, the accuracy and precision statistics given for other sensors should be for sensors at a similar price point to the sensors used in the paper. Vaisala CarboCap is not a comparable sensor (lines 64-68). Focus literature review on truly comparable low-cost NDIR sensors.**

Response:

Thank you very much for your valuable comments. We have modified the section of introduction in lines 44-52. We added a table (Table 1) to clearly compare the cost, accuracy, and deployment status (including key cities with dense networks) of different low- and mid-cost $CO_2$ sensors.

In addition, we have revised the grammar and expression issues suggested by the reviewer. We added a literature review on long-term drift correction methods and added a discussion on the limitations of the long-term co-location results and the suggested correction frequency. We have provided detailed responses to the comments in the following text.

Moreover, we think that the methodologies and accuracy on mid-cost sensors and networks can provide valuable references and comparison for this study.

Recently established high-density networks use both low-cost and mid-cost sensors, and although the sensors are low- or mid- cost, the final operation and maintenance of the networks are relative high, sensor cost is not so important compared with the total cost, but the precision and accuracy of the network matter, and some networks using LCS with good correction methods can achieve comparable precision with mid-cost sensors. For the low-cost sensors, examples like SenseAir LP8 $CO_2$ sensor network in Switzerland, and SenseAir SENSE-JJJ in Beijing. And for the mid-cost sensor networks, there are BEACON and Paris using Vaisala GMP343 and SenseAir HPP, respectively.

Moreover, both low-cost and mid-cost sensors operate on the NDIR principle, exhibiting significant environmental sensitivity, susceptibility to jumps, and long-term drifts. Therefore, whether it is low-cost or mid-cost, their calibration principles and methods are common. Based on our results, after environmental sensitivity correction and effective long-term drift calibration, the accuracy of low-cost sensors is comparable to that of mid-cost sensors.

Turnbull, J., DeCola, P., Mueller, K., Vogel, F., Karion, A., Lopez Coto, I. and Whetstone, J. (2022), IG3IS Urban Greenhouse Gas Emission Observation and Monitoring Best Research Practices, World Meteorological Organization Integrated Greenhouse Gas Information System, [online], https://ig3is.wmo.int/ (Accessed June 11, 2025)

**What is China's dual carbon goal (line 72)? Please add context for the international reader.**

Response: Thank you. It is that China aims at peaking carbon emissions before 2030 and achieving carbon neutrality before 2060. We have provided additional clarification regarding the China's Dual Carbon Goals.

**Line 95: is "homology" the intended word? Homogeneity maybe?**

Response:Yes, here "homology"is the intended word, and in environmental sciences, 'homology' refers to identical emission sources of air pollutants and $CO_2$ resulted from fossil fuel combustion.

**What exactly is meant by background noise level (line 113)?**

Response: We use the standard deviation of the signal to quantify the level of background noise or white noise (e.g. the amplitude of raw signal in Fig. 3a in the main text, blue dots). So, we used "standard deviation" to substitute "background noise level" according to your question. We have refined the text to enhance clarity.

**Figure 4 map labels are too small to be legible and is difficult to locate for readers unfamiliar with Beijing city. Also the latitude labels are cut off on the left side.**

Response: Thank you for this question. We added the $5^{th}$ and $6^{th}$ Ring in the map to show the location. But the base map provided by ESRI map server cannot display street and area names more clearly. We added an eagle-eye view map to show the site's location. We have addressed the issue of cut off label display in the map.

**Line 189-190 is confusing and grammatically incorrect. Same with like 193-194. Why can you say that hanging on an open window is the same a field-deployment? If the instruments are even partially indoors surely they are more temperature controlled than a true field deployment?**
**What is the recommended length of co-location with reference instrument for determining the correction coefficients? It would be helpful to include this in addition to the recommendation of a 3-month calibration interval.**

Response: Thank you for this question. Due to the limitation of experimental conditions, we set the instruments at the edge of the open window to directly take in the outside air. We compared the observed environmental conditions of indoor, our set-up instruments, and out door field in the following figure 1 (Also added in Fig. S8). We show that the instruments shared almost the same changing environments as the out door air conditions. The temperature mainly ranged from 0-40 ℃, and the RH mainly ranged from 0-60%. And both if them are quite different from the indoor conditions. We added these discussions in lines 222-232.

We have added the discussion of the recommended length of co-location with reference instrument for determining the correction coefficients in lines 131-141. It

was generally one week period of co-location with reference instrument for controlled environmental experiments to determine the environmental correction coefficients of the sensors, and another week co-location for post-quality check. And to fully understand the characteristics of long-term drift, at least one year is recommended to include the seasonal cycle (both summer and winter).

[Figure]

Figure 1. The temperature and humidity variation of LCS instruments under (a) indoor, (b) beside the window, and (c) field conditions during the experimental period.

**Line 296: "manufacture" -> "manufacturer"**
Response:Thank you for this comment. We revised accordingly.

**Line 324-325: missing spaces around ±**
Response:Thank you for this comment. We revised all such errors in the text accordingly.

**Please add additional discussion of the limitations of this study and how the findings may or may not translate to other LCS and environments. Are the recommendations made only for K30s?**
Response:
Thank you for the valuable suggestions. We have added the discussions for this issue in section 7 lines 365-374.

"Long-term drift is typically observed in low- and mid-cost NDIR $CO_2$ sensors. While our findings provide valuable references for K30 sensors, the recommended calibration frequency and observed seasonal cycle characteristics may be also applicable to other similar NDIR sensors, and the calibration method we used is universal and helpful in long-term drift corrections.

For other NDIR sensors, we recommend selecting several samples to conduct co-location with high-precision instrument under field conditions for at least one year to study their full characteristics. This extended co-location period is essential to comprehensively characterize both long-term drifts and the seasonal drift cycle. The results will provide critical guidance for remote calibration of high-density networks using this type of sensor. Furthermore, another practical substitute method is using standard gas, and the calibration frequency from the experience of this study is recommended at least one-three month."

**Have the authors explored alternatives to a co-location for drift correction every 3-6 months? This may not be feasible for large deployments. Did the authors explore the performance of a remote calibration strategy at all? How would the proposed calibration scale (in cost/time) to, say, 100s of sensors deployed?**

**Please add additional literature review describing the existing methods for LCS calibration.**

Response:

Thank you for these valuable suggestions. We added discussions in lines 65-80. Indeed, this may not be feasible for large deployments, it is useful to understand the sensor performance and characteristics in long-term drifts, and suggest us the calibration frequency after deployment. We also explored several other calibration methods for drifts. One robust and commonly used method is automatic/manual standard gas calibration. We used this standard gas method in the Beijing and Jinan networks with more than 160 instruments, the calibration frequency is one week for automatic standard gas, and 1-3 months for manual calibration, and the results showed mean biases -1.28 ~ -0.64 ppm with standard gas at one month scale (Cai et al., 2024) (below figure). The manual standard gas calibration is useful in mobile observations such as on-road observations using vehicles and vertical profile observations using tethered balloons (Liu et al., 2021; Bao et al., 2020).

[Figure]

Figure 2: Comparison of $CO_2$ concentrations measured by SENSE-IAP and the $CO_2$ concentrations of standard gas at Beijing-CNEMC site in January and February 2022. (a) The time series of $CO_2$ per minute in the whole measurement period; the blue dashed lines mark one hour of the standard gas measurement per week, and the green dashed lines mark the concentration of standard gas. (b) The points are the hourly means of values during one hour of standard gas measurement per week, with mean biases -1.28 ~ -0.64 ppm and RMSE 1.16 ~ 2.26 ppm.

We also explored a remote calibration strategy under specific weather conditions by using reference concentrations from models, with a relative lower accuracy than the standard gas or reference instrument methods, whose general model-observation mismatch is 0-5ppm (below figure 3). The preliminary results were published in BAMS, showing the $CO_2$ concentration gradients observed in Beijing (Han et al., 2024). More results on the accuracy of the Beijing networks are in preparation and to be submitted in the following studies.

[Figure]

Figure 3. Model and Picarro mismatch at some favorable weather conditions and urban Beijing (IAP), suburban Beijing (Xianghe, XH), and background Beijing (Shangdianzi, SDZ, and Xinglong, XL) sites.

The calibration cost using automatic standard gas (1 week frequency) can reach $300/station/year, which consumes 2 tanks of 8L 10 MPa gas (one tank work standard gas, and one tank target/quality-check standard gas). And the time cost is maintaining workers calibrating gas and sending the calibrated gas to the stations, generally 3-5 stations/day/worker depending on the distances of the stations from the laboratory. And thus 100s of sensors deployed will cost $30,000/year, and 1 worker of 1-2 month time.

We added a literature review of the long-term drift calibration for LCS networks in Switzerland, Beijing, California, and Paris, lines 58-81 in section 1.

**The data availability statement in my opinion does not follow the best practice for open research. Why have the authors not made their data and calibration codes readily available to all readers in an online repository?**

Response: Thank you for this question. We provided all the Picarro and SENSE-IAP data used in this paper (publicly available at https://doi.org/10.6084/m9.figshare.29310890.v3). We further added more descriptions on the calibration algorithm and procedures in the main text (lines 133-141), and we disclosed our calibration method as much as we can, and several similar methods and precision were achieved in other studies (Lian et al., 2024; Shusterman et al., 2016). However, due to the commercial confidential requirements (Beijing, Jinan, Fuzhou, and several other cities' LCS networks we served), the detailed calculation formulas cannot be disclosed in this research paper.

Bao, Z., Han, P., Zeng, N., et al., 2020. Observation and modeling of vertical carbon dioxide distribution in a heavily polluted suburban environment. Atmospheric and Oceanic Science Letters, 1-9.

Han, P., Yao, B., Cai, Q., et al., 2024. Support Carbon Neutral Goal with a High-Resolution Carbon Monitoring System in Beijing. Bulletin of the American Meteorological Society 105 (12), E2461-E2481.

Lian, J., Laurent, O., Chariot, M., et al., 2024. Development and deployment of a mid-cost CO2 sensor monitoring network to support atmospheric inverse modeling for quantifying urban CO2 emissions in Paris. Atmos. Meas. Tech. 17 (19), 5821-5839.

Liu, D., Sun, W., Zeng, N., et al., 2021. Observed decreases in on-road CO2 concentrations in Beijing during COVID-19 restrictions. Atmospheric Chemistry and Physics 21 (6), 4599-4614.

Martin, C.R., Zeng, N., Karion, A., et al., 2017. Evaluation and environmental correction of ambient CO(2) measurements from a low-cost NDIR sensor. Atmospheric measurement techniques 10.

Shusterman, A.A., Teige, V.E., Turner, A.J., et al., 2016. The BErkeley Atmospheric $CO_2$ Observation Network: initial evaluation. Atmos. Chem. Phys. 16 (21), 13449-13463.